# A dose-dependent beneficial effect of methotrexate on the risk of interstitial lung disease in rheumatoid arthritis patients

Joanna Kur-Zalewska[1,2]* , Bartłomiej Kisiel[1,2], Marta Kania-Pudło[3‡], Małgorzata Tłustochowicz[1‡], Andrzej Chciałowski[4‡], Witold Tłustochowicz[1‡]

1 Department of Internal Diseases and Rheumatology, Military Institute of Medicine, Warsaw, Poland, 2 Clinical Research Support Center, Military Institute of Medicine, Warsaw, Poland, 3 Department of Radiology, Military Institute of Medicine, Warsaw, Poland, 4 Department of Infectious Diseases and Allergology, Military Institute of Medicine, Warsaw, Poland

☯ These authors contributed equally to this work.
‡ These authors also contributed equally to this work.
* jkur-zalewska@wim.mil.pl

**Data Availability Statement:** All relevant data are within the manuscript and its Supporting Information files.

## Abstract

### Objectives

The aim of the study was to assess the influence of different factors, including treatment, on the risk of ILD in the course of RA.

### Methods

A total of 109 RA patients were included in the analysis. High-resolution computed tomography (HRCT) of chest was obtained in each patient. Patients were classified as having ILD (ILD group) or not (N-ILD group). The ILD was graded using the semi-quantitative Warrick scale of fibrosis. Warrick extent score (WES) and Warrick severity score (WSS) were calculated separately for each patient, then combined to obtain a global score (WGS).

### Results

In univariate analysis the presence of ILD was associated positively with age ($P = 5 \times 10^{-6}$) and negatively with MTX treatment ($P = 0.0013$), mean MTX dose per year of treatment ($P = 0.003$) and number of DMARDs used ($P = 0.046$). On multivariate analysis only age and treatment with MTX were independently associated with the presence of ILD. WGS was significantly lower in patients treated with MTX in a dose of $\geq 15$ mg/week (MTX$\geq$15 group) as compared to patients treated with lower doses of MTX (0<MTX<15 group) or not treated with MTX (N-MTX group) ($P = 0.04$ and $P = 0.037$, respectively). The ILD prevalence was higher in N-MTX group than in 0<MTX<15 group ($P = 0.0036$) and MTX$\geq$15 group (0.0007). The difference in ILD prevalence between MTX$\geq$15 and 0<MTX<15 groups was not significant, but the latter group had higher WES ($P = 0.044$) and trended to have higher WSS and WGS.

**Funding:** The authors received no specific funding for this work.

**Competing interests:** The authors have declared that no competing interests exist.

## Consclusions

We found a beneficial effect of MTX on RA-ILD. Importantly, this effect seems to be dose dependent.

## Introduction

Rheumatoid arthritis (RA) is a chronic systemic inflammatory disease manifested by articular and extra-articular features. Pulmonary involvement belongs to the most common extra-articular manifestation of RA, being detected in 40–70% of patients, and includes pleural lesions and effusions, rheumatoid nodules, interstitial lung disease (rheumatoid arthritis-associated interstitial lung disease–RA-ILD), obstructive lung disease, bronchiectasis, bronchiolitis, obliterative bronchiolitis as well as drug induced toxicity and opportunistic infections in the setting of immunosuppression [1–7].

RA-ILD is one of the most important RA complications because of its high morbidity and mortality. However, the range of severity of RA-ILD is broad with subtle or even non-existing symptoms at onset (despite radiographic features consistent with ILD). The 1-year incidence of RA-ILD has been reported at 2.8% [8]. The prevalence of ILD in patients with RA varies from 5% to 58%, depending on the techniques and criteria used to diagnose ILD as well as RA severity in the population studied [9–11]. Risk factors for RA-ILD include age, RA duration and older age at disease onset, male sex, high levels of rheumatoid factor (RF) and anti-cyclic citrullinated peptide antibodies (ACPA), cigarette smoking, elevated erythrocyte sedimentation rate (ESR), high disease activity (based on disease activity score 28- DAS28) and severity (worse functional status, presence of destructive changes and rheumatoid nodules) [12–15].

Importantly, ILD in RA patients may be associated not only with the disease itself (RA-ILD), but also with the drugs used for its treatment (drug-induced ILD–DI-ILD). DI-ILD has been reported to associate with all disease-modifying anti-rheumatic drugs (DMARDs), including methotrexate (MTX), leflunomide and biologics [16–31]. There are some clinical and radiological features that can helps differentiate DI-ILD from RA-ILD, including usually acute or subacute course of the DI-ILD characterized by cough, dyspnoe and fever and radiologic findings consistent with non-specific interstitial pneumonia, less often with bronchiolitis obliterans organizing pneumonia [32]. However, in clinical practice it is sometimes difficult to differentiate RA-ILD from DI-ILD given that clinical, radiological, and histopathological findings are non-specific and overlap [33]. The situation is additionally complicated by the fact that some patients may have discreet ILD changes (invisible in routine chest X-ray) prior to starting DMARDs therapy. Moreover, establishment of a causal relationship between a DMARD and ILD may be difficult because a large proportion of patients are treated with more than one DMARD (sequentially or in a combination therapy). In conclusion, there is still a debate about real risk of ILD related to use of DMARDs. The answer to this question is of huge importance as discontinuation of DMARDs therapy or even less aggressive treatment can worsen long-term prognosis and, paradoxically, increase the risk of RA-ILD and other extra-articular manifestations.

Although surgical lung biopsy is a gold standard for diagnosis of ILD, it is rarely used in clinical practice due to the risk associated with this procedure. High-resolution computed tomography (HRCT) of chest has been accepted as the gold standard noninvasive imaging method in the diagnosis of ILD in patients with RA [34]. HRCT results have been shown to correlate with the results of open lung biopsy [35, 36].

The aim of our study was to assess the association of different factors (with particular emphasis on the treatment with DMARDs and glucocorticosteroids) on the risk of ILD in the course of RA.

## Materials and methods

The study was approved by the Military Institute of Medicine Ethics Committee (69/WIM/2011). Written informed consent was obtained from each patient. All procedures were performed in accordance with the Helsinki Declaration of 1975, revised in 1983.

### Patients and methods

A consecutive RA patients hospitalized in the Department of Internal Medicine and Rheumatology, Military Institute of Medicine in Warsaw between December 2006 and April 2010, fulfilling inclusion and exclusion criteria, were recruited to the study. The inclusion criteria were as follows: (i) the diagnosis of RA established on the basis of 1987 ACR criteria; (ii) age ≥18 years; (iii) written informed consent to the study. The exclusion criteria included the following: (i) presence of malignancy; (ii) history of pulmonary thromboembolism; (iii) severe left ventricular failure; (iv) pneumonia; (v) previous diagnosis of ILD. The reason to exclude patients with pre-existing ILD was to avoid channeling bias, as the presence of this complication could affect choice of RA treatment regimen (i.e. avoidance of MTX and preferring other drugs, such as leflunomide) and our main goal was to assess the influence of DMARDs on ILD risk.

A structured questionnaire was used to collect data regarding age, gender, smoking habits, body mass index (BMI) and previous treatment S1 and S2 Files. All conventional and biologic DMARDs and glucocorticosteroids (GCS) used for at least 3 months were reported. Based on treatment history patients were divided into group treated (present and in the past) and not-treated with an individual medication. Additionally, on the basis of history of MTX treatment patients were divided into three groups: i) patients not treated with MTX (N-MTX group), ii) patients treated with a mean MTX dose<15 mg/week (0<MTX<15 group), iii) patients treated with a mean MTX dose ≥15 mg/week (MTX≥15 group).

The laboratory work-up included: erythrocyte sedimentation rate (ESR), C-reactive protein (CRP), rheumatoid factor (RF) and anti-citrullinated protein antibodies (ACPA). In order to assess disease activity and physical dysfunction DAS28 (disease activity score 28) and HAQ (health assessment questionnaire) were used, respectively.

HRCT of chest was performed in each patient included in the study. HRCT was scored by an experienced radiologist, who was aware of the RA diagnosis but was blinded to the clinical and laboratory data. ILD was diagnosed based on typical tomography pattern, defined as presence of diffuse peripheral or subpleural reticulation with or without honeycombing and ground-glass opacities [2, 3, 37]. On the basis of HRCT patients were classified as having ILD (ILD group) or not (N-ILD group). The ILD was graded using the semi-quantitative Warrick scale of fibrosis from 0 to 30 [37]. Warrick extent score (WES) and Warrick severity score (WSS) were calculated separately for each patient, then combined to obtain a global score (WGS). ILD was classified as mild (WGS <8 points), moderate (WGS 8–15 points) and severe (WGS >15 points), similarly as in the study by Fessi et al. [37].

### Statistical analysis

The statistical analyses were performed using Statistica 12 package (StatSoft Inc). Results are reported as mean (SD) for continuous variables and *n* (%) for categorical variables. The t-test and ANOVA were used to compare the means of two groups (t-test) and three or more groups

(ANOVA). Categorical variables were compared with chi square exact test. A logistic regression was used to predict the factors independently associated with ILD. A *P* value < 0.05 was considered significant.

## Results

Initially, 111 patients with RA were included into the study (95 women and 16 men). The mean age was 60.5±11.2 years (range 31–84 years). The mean RA duration was 9.94±9.42 years (0.25–40 years). Two patients were excluded from the analysis because of incomplete medical data concerning previous RA treatment. Eventually, a total of 109 patients were included in the analysis. The demographic and clinical characteristics of the patients are presented in Table 1.

Approximately 70% of patients had ILD on HRCT but most of them had mild changes (WGS <8); 12 (11.01%) and 10 (9.17%) patients had moderate and severe ILD, respectively (Table 2).

**Table 1. Study group characteristics.**

|  | Mean (SD) or n (%) |
|---|---|
| Age [years] | 60.50 (11.20) |
| Male sex | 16 (14.68%) |
| RA duration [years] | 9.94 (9.42) |
| DAS28 at the time of inclusion to the study | 5.19 (1.43) |
| HAQ | 1.88 (0.74) |
| RF [IU/mL] | 290.57 (559.00)[a] |
| Presence of RF | 83 (77.57%)[a] |
| ACPA [U/mL] | 248.92 (245.09)[b] |
| Presence of ACPA | 86 (82.69)[b] |
| Serum creatinine [mg/dl] | 0.745 (0.25) |
| Pack-years of smoking | 10.91 (14.62) |
| Smokers (current and in the past) | 60 (55%) |
| Treatment with MTX | 86 (78.90%) |
| Duration of MTX treatment [months] | 25.86 (31.76) |
| Cumulative dose of MTX [mg] | 1471.74 (1868.73) |
| Mean MTX dose per year of treatment [mg/year] | 529.49 (329.68) |
| Mean MTX dose ≥15 mg/week (during whole treatment period) | 25 (22.94%) |
| Treatment with SSA | 52 (47.71%) |
| Treatment with CsA | 29 (26.61%) |
| Treatment with gold salts | 25 (22.94%) |
| Treatment with chloroquine | 30 (27.52%) |
| Treatment with leflunomide | 19 (17.43%) |
| Treatment with biologics (etanercept, adalimumab, infliximab, rituximab) | 16 (14.68%) |
| Treatment with GCS | 89 (81.65%) |
| No of DMARDs (conventional and biologic) | 2.57 (2.07) |

Data available for 109 patients unless otherwise stated.

ACPA–Anti-Citrullinated Protein Antibodies; CsA–cyclosporine; DAS28 –Disease Activity Score 28; GCS–glucocorticosteroids; DMARDs–disease-modifying anti-rheumatic drugs; HAQ–Health Assessment Questionnaire; MTX–methotrexate; RA–rheumatoid arthritis; RF–rheumatoid factor; SSA–sulfasalazine [a]Data available for 107 patients; [b]Data available for 104 patients.

**Table 2. HRCT of the chest results.**

|  | Mean (SD) or n (%) |
|---|---|
| Presence of ILD | 74 (67.89%) |
| Presence of ground-glass opacities | 10 (9.17%) |
| Presence of irregular pleural margin | 30 (27.52%) |
| Presence of septal or subpleural lines | 71 (65.14%) |
| Presence of honeycombing | 10 (9.17%) |
| Presence of subpleural cyst | 12 (11%) |
| Warrick's extent score (WES) | 1.83 (2.57) |
| Warrick's severity score (WSS) | 3.50 (3.90) |
| Warrick's global score (WGS) | 5.33 (6.32) |
| ILD grade |  |
| • no ILD/mild ILD (WGS<8) | 87 (79.82%) |
| • moderate ILD (8≤WGS≤15) | 12 (11.01%) |
| • severe ILD (WGS>15) | 10 (9.17%) |

Data available for 109 patients unless otherwise stated.

ILD–interstitial lung disease; WES–Warrick's extent score; WSS–Warrick's severity score; WGS–Warrick's global score.

In univariate analysis the presence of ILD was associated positively with age and negatively with MTX treatment, mean MTX dose per year of treatment and number of DMARDs used; marginal association was observed for male sex (Table 3).

Multivariate analysis including age, treatment with MTX, number of DMARDs used and male sex showed that the presence of ILD in RA patients significantly increased with age ($P = 6 \times 10^{-6}$, beta = 0.0167) and significantly decreased with MTX treatment ($P = 0.0015$, beta = -0.315). Similar results were found for analysis including age, mean MTX dose per year of treatment (included patients not treated with MTX), number of DMARDs used and male sex–positive association was observed for age ($P = 2 \times 10^{-5}$, beta = 0.0162) and negative for MTX dose per year of treatment ($P = 0.012$, beta = -0.0003).

In order to find out if there is an association between ILD and methotrexate dose we compared WES, WSS and WGS in N-MTX (n = 23), 0<MTX<15 (n = 61) and MTX≥15 (n = 25) groups. The comparison showed marginal associations (Table 4). However, in the post-hoc analysis WGS was significantly lower in the MTX≥15 group as compared to N-MTX group and 0<MTX<15 group ($P = 0.04$ and $P = 0.037$, respectively). Moreover, WES, WSS and WGS did not differ significantly in N-MTX and 0<MTX<15 groups ($P = 0.92$, $P = 0.56$ and $P = 0.69$, respectively).

In order to evaluate if the observation of higher WGS in 0<MTX<15 group as compared to MTX≥15 group was due to higher prevalence of ILD or rather its more severe character we assessed the frequency of ILD in these groups and also compared WES, WSS and WGS in ILD patients (i.e. after excluding N-ILD patients) from both groups. No significant difference was found in terms of ILD frequency (63.93% vs 52% for 0<MTX<15 and MTX≥15, respectively, P = 0.3). However, ILD patients from o<MTX<15 group had higher WES than ILD patients from MTX≥15 group (3.31±3.05 vs 1.54±0.66, $P = 0.044$) and trended to have higher WSS (5.95±4.20 vs 3.85±1.41, $P = 0.08$) and WGS (9.26±6.96 vs 5.38±1.94, $P = 0.054$).

It should be noted that although N-MTX and 0<MTX<15 groups did not differed significantly in terms of WES, WSS and WGS (these parameters were only slightly higher in N-MTX group), the prevalence of ILD was significantly higher among N-MTX patients (22/23–95.65%

**Table 3. Association of ILD with demographic and clinical characteristics of patients.**

| | ILD (n = 74) | N-ILD (n = 35) | *P* |
|---|---|---|---|
| Age [years] | 63.74 (8.40) | 53.66 (13.25) | **5 x 10⁻⁶** |
| Male sex | 14 (18.92%) | 2 (5.71%) | 0.07 |
| RA duration [years] | 9.86 (10.13) | 10.10 (7.82) | 0.90 |
| DAS28 at the time of inclusion to the study | 5.31 (1.51) | 4.95 (1.21) | 0.22 |
| HAQ | 1.85 (0.80) | 1.96 (0.61) | 0.46 |
| RF [IU/mL][a] | 349.90 (663.30) | 168.54 (180.70) | 0.12 |
| Presence of RF[a] | 58 (80.56%) | 25 (71.43%) | 0.29 |
| ACPA [U/mL][b] | 263.46 (246.79) | 216.19 (241.86) | 0.37 |
| Presence of ACPA[b] | 61 (84.72%) | 25 (78.13%) | 0.41 |
| Serum creatinine [mg/dl] | 0.774 (0.25) | 0.682 (0.25) | 0.075 |
| Presence of eGFR<60 ml/min/1.73m² | 9 (12.16%) | 4 (11.43%) | 0.9 |
| Pack-years of smoking | 11.47 (16.03) | 9.72 (11.16) | 0.56 |
| Treatment with MTX | 52 (70.27%) | 34 (97.14%) | **0.0013** |
| Duration of MTX treatment [months] | 23.07 (32.69) | 31.77 (29.26) | 0.18 |
| Cumulative dose of MTX [mg] | 1313.38 (1879.63) | 1806.57 (1826.78) | 0.20 |
| Mean MTX dose per year of treatment [mg/year] | 465.83 (349.75) | 664.09 (234.81) | **0.003** |
| Mean MTX dose ≥15 mg/week (during whole treatment period) | 13 (17.57%) | 12 (34.29%) | 0.052 |
| Treatment with SSA | 35 (47.30%) | 17 (48.57%) | 0.90 |
| Treatment with CsA | 17 (22.97%) | 12 (34.29%) | 0.18 |
| Treatment with gold salts | 15 (20.27%) | 10 (28.57%) | 0.30 |
| Treatment with chloroquine | 17 (22.97%) | 13 (37.14%) | 0.12 |
| Treatment with leflunomide | 12 (16.22%) | 7 (20.00%) | 0.63 |
| Treatment with biologics (etanercept, adalimumab, infliximab, rituximab) | 8 (10.81%) | 8 (22.86%) | 0.10 |
| Treatment with GCS | 58 (78.38%) | 31 (88.57%) | 0.20 |
| No of DMARDs (conventional and biologic) | 2.30 (1.88) | 3.14 (2.35) | **0.046** |

ACPA–Anti-Citrullinated Protein Antibodies; CsA–cyclosporine; DAS28 –Disease Activity Score 28; DMARDs–disease-modifying anti-rheumatic drugs; GCS–glucocorticosteroids; HAQ–Health Assessment Questionnaire; MTX–methotrexate; RA–rheumatoid arthritis; RF–rheumatoid factor; SSA–sulfasalazine

[a]Data available for 109 patients

[b]Data available for 106 patients.

vs 39/61–63.93%, *P* = 0.0036). The prevalence of ILD was also significantly higher in N-MTX group as compared to MTX≥15 group (*P* = 0.0007).

## Discussion

ILD is one of the most severe extra-articular manifestation of RA. As we mentioned above risk factors for the development of RA-ILD include: age, RA duration and older age at disease onset, male sex, high levels of RF and ACPA, cigarette smoking, high disease activity and severity. In our cohort among the above mentioned factors only age showed a clear association with ILD (in multivariate analysis). Lack of association between other factors and ILD in our study may be due to relatively small study group but also to the exclusion criteria used–patients with previously diagnosed ILD were excluded from the study. This, on one hand, reduces the risk of channeling bias and thus facilitates evaluation of the effects of drugs. On the other hand, the excluded group undoubtedly comprises a large proportion of patients with clinically severe

**Table 4. Comparison of Warrick score depending on methotrexate dose.**

| | Type of comparison | | | | |
| --- | --- | --- | --- | --- | --- |
| | NMTX vs 0<MTX<15 vs MTX≥15 | | | | |
| | NMTX | 0<MTX<15 | MTX≥15 | P | Post-hoc analysis |
| WES, mean (SD) | 2.17 (0.53) | 2.11 (0.32) | 0.80 (0.51) | 0.074 | NMTX vs 0<MTX<15 – P = 0.92 |
| | | | | | NMTX vs MTX≥15 – P = 0.063 |
| | | | | | 0<MTX<15 vs MTX≥15 – P = 0.031 |
| WSS, mean (SD) | 4.35 (0.80) | 3.80 (0.49) | 2.00 (0.77) | 0.075 | NMTX vs 0<MTX<15 – P = 0.56 |
| | | | | | NMTX vs MTX≥15 – P = 0.037 |
| | | | | | 0<MTX<15 vs MTX≥15 – P = 0.05 |
| WGS, mean (SD) | 6.52 (1.30) | 5.92 (0.80) | 2.80 (1.24) | 0.067 | NMTX vs 0<MTX<15 – P = 0.69 |
| | | | | | NMTX vs MTX≥15 – P = 0.04 |
| | | | | | 0<MTX<15 vs MTX≥15 – P = 0.037 |
| | MTX 0–15 vs MTX≥15 | | | | |
| | MTX 0–15 | | MTX≥15 | | P |
| WES, mean (SD) | 2.13 (0.28) | | 0.80 (0.50) | | 0.022 |
| WSS, mean (SD) | 3.95 (0.42) | | 2.00 (0.77) | | 0.027 |
| WGS, mean (SD) | 6.08 (0.68) | | 2.80 (1.24) | | 0.022 |

WES–Warrick extent score; WGS–Warrick global score; WSS–Warrick severity score; NMTX–no treatment with methotrexate; MTX 0–15 –no treatment with methotrexate or methotrexate treatment with a mean dose of <15 mg/ per year of treatment; 0<MTX<15 –methotrexate treatment with a mean dose of <15 mg/per year of treatment; MTX≥15 –methotrexate treatment with a mean dose of ≥15 mg/per year of treatment.

ILD and we may speculate that in these patients the influence of risk factors might be even more pronounced.

As high disease activity is one of the RA-ILD risk factor, a proper treatment with DMARDs should protect against its development. On the other hand ILD was reported after virtually all DMARDs, although the actual scale of this complication is difficult to assess. DI-ILD is one of the most feared complication of MTX. Some studies reported its prevalence to be as high as 11.6% [38]. However, according to more recent studies it seems to be far less frequent. In a large cohort study (673 patients, 1402 patient-years of treatment with MTX) Kinder et al. found only 6 cases (0.9%) of MTX pneumonitis [39]. A systematic literature research by Salliot and van der Heijde reported the prevalence of MTX pneumonitis in patients on long-term MTX monotherapy as 0.43% (15 cases among 3463 patients) [40]. The risk of drug-induced

pneumonitis for other DMARDs, such as leflunomide and biologics, seems to be similar as for MTX [41]. In regard of DMARDs, especially MTX, there are concerns regarding the possibility of inducing/exacerbating RA-ILD. However, a growing body of evidence suggests that there is no correlation between the use of MTX and the development or exacerbation of RA-ILD. Moreover, some studies shows a beneficial effect of MTX on RA-ILD risk and course. Two meta-analyses of randomized controlled trials by Conway et al. showed no evidence of association between MTX use and ILD in RA and non-RA inflammatory diseases [42, 43], while a study by Rojas-Serrano et al. found that patients who received MTX as part of RA-ILD treatment had a better survival than those not receiving MTX [44]. Similarly, in a study by Kiely et al. MTX treatment was not only not associated with an increased risk of RA-ILD diagnosis but instead it seemed to delay the onset of ILD [45]. The results of our study concur with the above-mentioned, suggesting that MTX decreases the RA-ILD risk. However, we also found that this beneficial effect of MTX was dose-dependent. Thus, it seems that the MTX dose determines not only its efficacy in controlling articular symptoms but also in prevention of extra-articular complications. It should be emphasized that the treatment with MTX is effective, safe, and relatively cheap. Moreover, MTX was shown to reduce mortality in RA [46]. Therefore, taking in consideration the benefits of MTX treatment, a decision to withhold it as a treatment option for RA should be made only for sound reasons. It is suggested by some authors that MTX should only be withheld from RA patients with insufficient respiratory reserve to make it unlikely that they would survive hypersensitivity pneumonitis [45]. Our study, showing a beneficial effect of MTX on RA-ILD, support this point of view. On the other hand, even though drug induced pneumonitis is a relatively rare complication, one should always have in mind this possibility, especially in patients with risk factors.

Although we did not find any significant association between other (than MTX) DMARDs and the risk of ILD, it should be noted that the percentage of patients treated with biologics (22.86% vs 10.81%, $P = 0.1$), chloroquine (37.14% vs 22.97%, $P = 0.12$) and cyclosporine (34.29% vs 22.97%, $P = 0.18$) was higher in N-ILD group as compared to ILD group. Thus, it is possible that other DMARDs exert a similar influence on RA-ILD as MTX and lack of association was simply due to relatively small study group.

The study has several limitations. First, it was an observational study, therefore the channeling bias cannot be ruled out. However, this bias should be of limited importance as patients with already diagnosed ILD were not included in the study. Second, the study involved only in-patients, thus, we may speculate that our study group comprised patients with "more severe disease" and the results may not be fully representative of the whole RA population. Third, the study group was relatively small. Therefore, our results should be confirmed by studies on larger cohorts.

## Conclusions

We found a beneficial effect of MTX on RA-ILD. Importantly, this effect seems to be dose dependent. Further studies on larger cohorts are warranted to confirm our results.

## Supporting information

**S1 File. Kwestionariusz pacjenta.**
(PDF)

**S2 File. Questionnaire.**
(PDF)

## Author Contributions

**Conceptualization:** Joanna Kur-Zalewska, Andrzej Chciałowski, Witold Tłustochowicz.

**Data curation:** Joanna Kur-Zalewska, Marta Kania-Pudło, Małgorzata Tłustochowicz.

**Formal analysis:** Joanna Kur-Zalewska, Bartłomiej Kisiel, Andrzej Chciałowski, Witold Tłustochowicz.

**Investigation:** Joanna Kur-Zalewska.

**Methodology:** Joanna Kur-Zalewska, Bartłomiej Kisiel, Andrzej Chciałowski, Witold Tłustochowicz.

**Project administration:** Joanna Kur-Zalewska.

**Supervision:** Witold Tłustochowicz.

**Writing – original draft:** Joanna Kur-Zalewska, Bartłomiej Kisiel.

**Writing – review & editing:** Joanna Kur-Zalewska, Bartłomiej Kisiel, Marta Kania-Pudło, Małgorzata Tłustochowicz, Andrzej Chciałowski, Witold Tłustochowicz.

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
