## [Decision Letter · Decision Letter 0]

18 Dec 2020

PONE-D-20-32723

A dose-dependent beneficial effect of methotrexate on the risk of interstitial lung disease in rheumatoid arthritis patients.

PLOS ONE

Dear Dr. Kur-Zalewska

Thank you for submitting your manuscript to PLOS ONE. After careful consideration, we feel that it has merit but does not fully meet PLOS ONE’s publication criteria as it currently stands. Therefore, we invite you to submit a revised version of the manuscript that addresses the points raised during the review process.

We look forward to receiving your revised manuscript.

Kind regards,

Minghua Wu, M.D., Ph.D.

Academic Editor

PLOS ONE

2. Please ensure that you include a title page within your main document.

We do appreciate that you have a title page document uploaded as a separate file, however, as per our author guidelines (http://journals.plos.org/plosone/s/submission-guidelines#loc-title-page) we do require this to be part of the manuscript file itself and not uploaded separately.

3. Please provide additional details regarding participant consent.

In the ethics statement in the Methods and online submission information, please ensure that you have specified what type you obtained (for instance, written or verbal, and if verbal, how it was documented and witnessed). If your study included minors, state whether you obtained consent from parents or guardians.

If the need for consent was waived by the ethics committee, please include this information.

4. Please include additional information regarding the survey or questionnaire used in the study and ensure that you have provided sufficient details that others could replicate the analyses.

For instance, if you developed a questionnaire as part of this study and it is not under a copyright more restrictive than CC-BY, please include a copy, in both the original language and English, as Supporting Information.

5. In your Methods section, please provide additional information about the participant recruitment method and the demographic details of your participants. Please ensure you have provided sufficient details to replicate the analyses such as:

a) the recruitment date range (month and year),

b) a statement as to whether your sample can be considered representative of a larger population, and

c) a description of how participants were recruited.

7. Please amend your manuscript to include your abstract after the title page.

Reviewers' comments:

Reviewer's Responses to Questions

**Comments to the Author**

1. Is the manuscript technically sound, and do the data support the conclusions?

Reviewer #1: Partly

Reviewer #2: Yes

2. Has the statistical analysis been performed appropriately and rigorously? 

Reviewer #1: Yes

Reviewer #2: Yes

3. Have the authors made all data underlying the findings in their manuscript fully available?

Reviewer #1: No

Reviewer #2: Yes

4. Is the manuscript presented in an intelligible fashion and written in standard English?

Reviewer #1: Yes

Reviewer #2: Yes

5. Review Comments to the Author

Reviewer #1: This is a really interesting study. However, there are several limitations mainly pertain to the way that the results are presented. Also, there are many methodological issues that need to be addressed. Please find my specific comments bellow.

Major comments

Patients and controls: firstly, I would change this subtitle to “patients and methods”. Besides, cannot see any control group. Can the authors clarify during which period these patients were recruited? Also, I assume this was a prospective study. Is that so?

Patients and controls: could the authors please clarify what “incomplete medical data” means?

Patients and controls: Who scored the HRCTs? Was that a radiologist? If so, was she/he blinded to the clinical/lab assessment?

Patients and controls: It would be useful to know how many patients were excluded due to other reasons (e.g pre-existing ILD).

Patients and controls: If I understand correctly, a HRCT was performed in every RA patient admitted in the hospital (“in-patients”). If so, is that in accordance with the ethical approval? Also, for what reason these patients were initially admitted?

Table-1

DAS28. Was this at the time of inclusion to the study?

pack-years of smoking. Useful information. However, how many patients were smokers?

No of DMARDS: do the authors mean biologic/conventional or both?

Biologics: which biologics were they?

GCS: is it available the mean/cumulative dose as well as duration of treatment?

I would be tempted to break table-1 in to two tables. The second one could present the radiologic characteristics/scoring.

Results: was treatment with MTX associated with the occurrence of ILD? Or was it protective for this event? I am saying that because in the univariate analysis, it seems that patients treated with MTX have more chances NOT to have ILD. However, in the multivariable analysis, it is implied the opposite. Said that, can we have some more details for the multivariate analysis (i.e exp(B) etc….).

Results: In this section, it appears for the first time that patients were divided according to the dose of MTX used. This should appear in the methods section.

Additionally, I find the section of comparison between groups with different doses of MTX, extremely difficult to follow. It needs to be re-written in a simpler way. The authors might consider to break that down into smaller paragraphs, with a concluding sentence in each of them. Also, have the authors checked if cumulative dose of MTX could affect the occurrence of ILD?

Discussion: if a difference does not reach the level of statistical significance, then it’s not a difference. Therefore, I would avoid to report that (e.g male gender, smoking) in the discussion.

Minor comments

Introduction: “…The occurrence of DI-ILD was reported after virtually all disease-modifying anti-rheumatic drugs..”. This sentence needs rephrasing. I would change that to “DI-ILD has been reported to associate with all disease-modifying anti-rheumatic drugs”

Introduction: I am not sure that “discrete” is the right word. “occult” or similar might suit better.

Introduction: I slightly disagree with the sentence that the distinction between RA-ILD and DI-ILD is difficult in clinical grounds. Although, these two can overlap and sometimes additional tests have to be performed, there are some clinical/radiological and other features (reviewed here: Fragoulis GE et al, Front in Med 2019) that can help in the differential diagnosis. I would add a sentence, to capture that.

Introduction: aim of the study: I would change “influence” to “association”.

Informed consent: was that written. Please clarify.

Patients and controls: “this complication would, with great probability, influence RA treatment regimen”. I would change that to “…this complication could affect choice of treatment regime”

Table-1: for convenience I would put the total number of patients included in the heading/subheading of the table.

Table-1: QA does not correspond anywhere. Same applies for table-2.

Reviewer #2: Joanna Kur-Zalewska and colleagues assessed the effect of methotrexate on the risk of interstitial lung disease in patients with rheumatoid arthritis. This report is valuable for clinical daily practice and well-described importance of MTX in patients with ILD in the discussion. However, there are several comments.

Major issues that need clarification:

#1. On Method

Did you collect the data on HRCT retrospectively?

If so, I concern that Warrick’s score in patients with infectious pneumonia would be greater than the score in patients without pneumonia. Thus, please answer questions as follow,

#1-1. Did you exclude the patients with pneumonia?

#1-2. Could you clarify the reason for hospitalization and the reason for performing a CT scan?

#2. On Method

Why did you include only in-patients? As you mentioned in the discussion, this restriction leads to the selection bias

#3. On Method

You should clarify the definition of ILD. Warrick’s score is not a criterion of ILD.

#4. On Method

How many readers scored Warrick’s score? And if more than two readers scored, how did they deal with any discrepancies between readers?

#5. On Method and Table 1

You should explain the definition of “treatment with MTX” and with other treatment agents in more detail. Do these terms mean only for current use or do they permit previous use within several years?

#6. On Table 1

Could you show the percentage of each item (Ground-glass opacities, Irregular pleural margin, Septal or subpleural lines, Honeycombing, and Subpleural cyst) in Warrick’s severity score? it would be helpful for the reader to consider the certainty of ILD in this study.

#7. On Table 2

The patients with ILD are elder than those with non-ILD. So, is there any possibility that the difference in chronic kidney disease prevalence affects the difference in MTX use ratio? In either case, could you add data on serum creatinine into Table 2 ?

6. PLOS authors have the option to publish the peer review history of their article (what does this mean?). If published, this will include your full peer review and any attached files.

Reviewer #1: **Yes: **George Fragoulis

Reviewer #2: **Yes: **Suguru Honda

---

## [Author Response · Author response to Decision Letter 0]

1 Mar 2021

Response to Journal requirements

1. Comment: Please ensure that your manuscript meets PLOS ONE's style requirements, including those for file naming. The PLOS ONE style templates can be found at https://journals.plos.org/plosone/s/file?id=wjVg/PLOSOne_formatting_sample_main_body.pdf and https://journals.plos.org/plosone/s/file?id=ba62/PLOSOne_formatting_sample_title_authors_affiliations.pdf

Response: The appropriate corrections according to PLOS ONE’s style requirements were done. 

2. Comment: Please ensure that you include a title page within your main document.

We do appreciate that you have a title page document uploaded as a separate file, however, as per our author guidelines (http://journals.plos.org/plosone/s/submission-guidelines#loc-title-page) we do require this to be part of the manuscript file itself and not uploaded separately. 

Response: We included the title page into the beginning of our manuscript file itself, listing all authors and affiliations. 

3. Comment: Please provide additional details regarding participant consent.

In the ethics statement in the Methods and online submission information, please ensure that you have specified what type you obtained (for instance, written or verbal, and if verbal, how it was documented and witnessed). If your study included minors, state whether you obtained consent from parents or guardians. If the need for consent was waived by the ethics committee, please include this information.

Response: Patients gave written informed consent. The appriopriate paragraph and online submission information were changed. Our study included only adults. 

4. Comment: Please include additional information regarding the survey or questionnaire used in the study and ensure that you have provided sufficient details that others could replicate the analyses. For instance, if you developed a questionnaire as part of this study and it is not under a copyright more restrictive than CC-BY, please include a copy, in both the original language and English, as Supporting Information.

Response: We included a copy of questionnaire used in our study, in both the original and English language, as Supporting Information. 

5. Comment: In your Methods section, please provide additional information about the participant recruitment method and the demographic details of your participants. Please ensure you have provided sufficient details to replicate the analyses such as:

a) the recruitment date range (month and year),

b) a statement as to whether your sample can be considered representative of a larger population, and

c) a description of how participants were recruited.

 Response: A consecutive RA patients, hospitalized in Department of Internal Medicine and Rheumatology Military Institute of Medicine in Warsaw between December 2006 and April 2010, fulfilling inclusion and exclusion criteria, were recruited to the study.

We provided additional information about the patient recruitment date and method in the Method section. We also provided additional demographic details of our participants in the Results section. 

We recruited to the study only in-patients and excluded patients with important co-morbidities (e.g. pulmonary thromboembolism, pneumonia, pre-existing interstitial lung disease, left ventricular failure and malignancies). That is why our sample can’t be considered representative of a general RA population.

6. Comment: PLOS requires an ORCID iD for the corresponding author in Editorial Manager on papers submitted after December 6th, 2016. Please ensure that you have an ORCID iD and that it is validated in Editorial Manager. To do this, go to ‘Update my Information’ (in the upper left-hand corner of the main menu), and click on the Fetch/Validate link next to the ORCID field. This will take you to the ORCID site and allow you to create a new iD or authenticate a pre-existing iD in Editorial Manager. Please see the following video for instructions on linking an ORCID iD to your Editorial Manager account: https://www.youtube.com/watch?v=_xcclfuvtxQ

 Response: My ORCID ID is validated in Editorial Manager. 

7. Comment: Please amend your manuscript to include your abstract after the title page.

Response: We included abstract after the title page. 

Response to Reviewer #1 comments

1. Comment: Patients and controls: firstly, I would change this subtitle to “patients and methods”. Besides, cannot see any control group.

Response: A subtitle „Patients and controls“ was changed to „Patients and method“.

2. Comment: Can the authors clarify during which period these patients were recruited? Also, I assume this was a prospective study. Is that so?

Response: It was a prospective study. Patients were recruited from Decemeber 2006 to April 2010. The appropriate paragraph was changed. 

3. Comment: Patients and controls: could the authors please clarify what “incomplete medical data” means?

Response: Two patients was excluded from analysis due to incomplete medical data concerning previous RA treatment. The appropriate paragraph was changed.

4. Comment: Patients and controls: Who scored the HRCTs? Was that a radiologist? If so, was she/he blinded to the clinical/lab assessment?

Response: HRCT was scored by an experienced radiologist (MK-P). The radiologist was aware of the main diagnosis (RA) but was blinded to the clinical state of the patients and their laboratory tests results. The appropriate paragraph was changed. 

5. Comment: Patients and controls: It would be useful to know how many patients were excluded due to other reasons (e.g pre-existing ILD).

Response: We agree that it would be useful to know how many patients were excluded due to other reasons (e.g. pre-existing ILD, pneumonia, lack of consent), but we haven’t registered patients excluded from the study. 

6. Comment: Patients and controls: If I understand correctly, a HRCT was performed in every RA patient admitted in the hospital (“in-patients”). If so, is that in accordance with the ethical approval? Also, for what reason these patients were initially admitted?

Response: HRCT of the lung was performed in every RA patient fulfilling inclusion and exclusion criteria to the study, including written informed consent. The study protocol was approved by a local ethics committee. It should be emphasized that ILD is a relatively common complication of RA, usually undectable in its early stages by a chest X-ray. Thus the only way to accurately assess the frequency and severity of ILD in RA patients is to perform HRCT.

Patients were admitted to the hospital mainly to perform a clinical checkups or because of a RA flare.

7. Table-1 

Comment: DAS28. Was this at the time of inclusion to the study?

Response: Yes. We added this information in Table1. 

Comment: Pack-years of smoking. Useful information. However, how many patients were smokers?

Response: Smoking status: current smokers- 18 patients, total (current and in the past)- 60 patients. We added this information in Table1. 

Comment: No of DMARDS: do the authors mean biologic/conventional or both?

Response: Both, conventional and biologic DMARDs. We added this information in Table1. 

Comment: Biologics: which biologics were they?

Response: Patients included into the study were treated with etanercept, adalimumab, infliximab and rituximab. We added this information in Table1. 

Comment: GCS: is it available the mean/cumulative dose as well as duration of treatment?

Response: Treatment with GCS in RA patients is variable over time. Calculation of cumulative GCS dose and duration of GCS treatment GCS is difficult and burdened with a large error, esspecially in patients with longstanding disease. In our study the mean RA duration was 9,94 years. Initially, we tried to estimate cumulative dose and time of GCS treatment based on medical history and medical records available, but there were too many doubts to obtain reliable, complete data. 

Comment: I would be tempted to break table-1 in to two tables. The second one could present the radiologic characteristics/scoring.

Response: We diveded Table 1 into two tables, first- Study group characteristic and second- HRCT of the chest results. 

8. Comment: Results: was treatment with MTX associated with the occurrence of ILD? Or was it protective for this event? I am saying that because in the univariate analysis, it seems that patients treated with MTX have more chances NOT to have ILD. However, in the multivariable analysis, it is implied the opposite. Said that, can we have some more details for the multivariate analysis (i.e exp(B) etc….).

Response: In our study, treatment with MTX was protective for ILD. It was confirmed by univariate and multivariate analysis- we refilled data of multivariate analysis. 

9. Comment: Results: In this section, it appears for the first time that patients were divided according to the dose of MTX used. This should appear in the methods section.

Response: The Method section was changed.

10. Comment: Additionally, I find the section of comparison between groups with different doses of MTX, extremely difficult to follow. It needs to be re-written in a simpler way. The authors might consider to break that down into smaller paragraphs, with a concluding sentence in each of them.

Response: The appropriate paragraph was changed. 

Aditionally, we corrected writers mistake in the section of comparison between groups with different doses of MTX. The prevalence of ILD was significantly higher in N-MTX group than in MTX≥15 group (22/23 – 95.65% vs 13/25 – 52%, P=0.0007). We changed wrong sentence placed in the original manuscript: The prevalence of ILD was also significantly higher in MTX≥15 group as compared to N-MTX group (P=0.0007) to the correct version: The prevalence of ILD was also significantly higher in N-MTX group as compared to MTX≥15 group (P=0.0007). 

11. Comment: Also, have the authors checked if cumulative dose of MTX could affect the occurrence of ILD?

Response: In our study, the cumulative MTX dose didn’t affect the occurence of ILD. The mean cumultive MTX dose was higher in N-ILD group (1806.57 mg) than in ILD group (1313.38 mg), but the difference wasn’t significant (p=0.20). 

12. Comment: Discussion: if a difference does not reach the level of statistical significance, then it’s not a difference. Therefore, I would avoid to report that (e.g male gender, smoking) in the discussion.

Response: The appropriate sentence was removed.

Minor comments

13. Comment: Introduction: “…The occurrence of DI-ILD was reported after virtually all disease-modifying anti-rheumatic drugs..”. This sentence needs rephrasing. I would change that to “DI-ILD has been reported to associate with all disease-modifying anti-rheumatic drugs”

Response: The sentence was changed.

14. Comment: Introduction: I am not sure that “discrete” is the right word. “occult” or similar might suit better.

Response: The appriopriate sentence was changed.

15. Comment: Introduction: I slightly disagree with the sentence that the distinction between RA-ILD and DI-ILD is difficult in clinical grounds. Although, these two can overlap and sometimes additional tests have to be performed, there are some clinical/radiological and other features (reviewed here: Fragoulis GE et al, Front in Med 2019) that can help in the differential diagnosis. I would add a sentence, to capture that.

Response: The appropriate paragraph was changed.

16. Comment: Introduction: aim of the study: I would change “influence” to “association”.

Response: The appriopriate sentence was changed.

17. Comment: Informed consent: was that written. Please clarify.

Response: It was written informed consent. The appropriate paragraph was changed.

18. Comment: Patients and controls: “this complication would, with great probability, influence RA treatment regimen”. I would change that to “…this complication could affect choice of treatment regime”

Response: The appriopriate sentence was changed.

19. Comment: Table-1: for convenience I would put the total number of patients included in the heading/subheading of the table.

Response: We added the total number of patients in the heading of the table. 

20. Comment: Table-1: QA does not correspond anywhere. Same applies for table-2.

Response: It was corrected.

Response to Reviewer #2 comments

1. Comment: On Method

Did you collect the data on HRCT retrospectively? If so, I concern that Warrick’s score in patients with infectious pneumonia would be greater than the score in patients without pneumonia. Thus, please answer questions as follow,

1-1. Did you exclude the patients with pneumonia?

Response: We excluded patients with pneumonia. The appriopriate paragraph was changed.

1-2. Could you clarify the reason for hospitalization and the reason for performing a CT scan?

Response: Patients were admitted to the hospital mainly to perform a clinical checkups or because of a RA flare. 

HRCT of the lung was performed in every RA patient fulfilling inclusion and exclusion criteria to the study, including written informed consent. The study protocol was approved by a local ethics committee. It should be emphasized that ILD is a relatively common complication of RA, usually undectable in its early stages by a chest X-ray. Thus the only way to accurately assess the frequency and severity of ILD in RA patients is to perform HRCT.

2. Comment: On Method

Why did you include only in-patients? As you mentioned in the discussion, this restriction leads to the selection bias

Response: We included into our study only in-patients due to administrative reasons. In our hospital, outpatient clinic is completely separate from the hospital departments (there is other staff employed, other procedures, different availability for diagnostic tests, include HRCT). There was no possibility to perform our study in outpatient clinic.

3. Comment: On Method

You should clarify the definition of ILD. Warrick’s score is not a criterion of ILD.

Response: ILD was diagnosed based on typical tomography pattern, defined as presence of diffuse peripheral or subpleural reticulation with or without honeycombing and ground-glass opacities. The appropriate paragraph wa changed.

4. Comment: On Method

How many readers scored Warrick’s score? And if more than two readers scored, how did they deal with any discrepancies between readers?

Response: HRCT was scored by an experienced radiologist, who was aware of the RA diagnosis but was blinded to the clinical and laboratory data. The appropriate paragraph was changed.

5. Comment: On Method and Table 1

You should explain the definition of “treatment with MTX” and with other treatment agents in more detail. Do these terms mean only for current use or do they permit previous use within several years?

Response: All conventional and biologic DMARDs and glucocorticosteroids (GCS) used for at least 3 months were recorded. Based on treatment history patients were divided into group treated (present and in the past) and not-treated with individual medication. We provided additional information in the Method section. 

6. Comment: On Table 1

Could you show the percentage of each item (Ground-glass opacities, Irregular pleural margin, Septal or subpleural lines, Honeycombing, and Subpleural cyst) in Warrick’s severity score? it would be helpful for the reader to consider the certainty of ILD in this study.

Response: Table 1 was divided into two parts: first- Study group characteristic and second- HRCT of the chest results. 

The presence of each Warrick‘s scale item (ground-glass opacities- 10 patients, irregular pleural margin- 30 patients, septal or subpleural lines- 71 patients, honeycombing- 10 patients, subpleural cyst- 12 patients) was added in Table 2. 

However, to clarify the diagnostic process it should be emphasized that the radiologic diagnosis of ILD was made by an experienced radiologist (based on typical tomography pattern, defined as presence of diffuse peripheral or subpleural reticulation with or without honeycombing and ground-glass opacities) and Warrick scale was just used to grade ILD (thus technically all patients with Warrick scale >0 had ILD).

7. Comment: On Table 2

The patients with ILD are elder than those with non-ILD. So, is there any possibility that the difference in chronic kidney disease prevalence affects the difference in MTX use ratio? In either case, could you add data on serum creatinine into Table 2 ?

Response: In our study, the mean serum creatinine was higher in ILD group (0.774 mg/dl) than in N-ILD group (0.682 mg/dl), but the difference wasn’t significant (p=0.075). Aditionally the percentage of patients with eGFR < 60 ml/min/1.73m2 was similar in both group (12% in ILD group vs 11.43% in N-ILD group, p=0.9). The difference in serum creatinine was probably associated with older age patients with ILD and didn‘t affect the use of MTX. We added serum creatinine in Table 1 and 3, we also added presence of eGFR < 60 ml/min/1.73m2 in Table 3.

---

## [Decision Letter · Decision Letter 1]

6 Apr 2021

A dose-dependent beneficial effect of methotrexate on the risk of interstitial lung disease in rheumatoid arthritis patients.

PONE-D-20-32723R1

Dear Dr. Zalewska,

We’re pleased to inform you that your manuscript has been judged scientifically suitable for publication and will be formally accepted for publication once it meets all outstanding technical requirements.

Kind regards,

Minghua Wu, M.D., Ph.D.

Academic Editor

PLOS ONE

Additional Editor Comments (optional):

Reviewers' comments:

Reviewer's Responses to Questions

**Comments to the Author**

1. If the authors have adequately addressed your comments raised in a previous round of review and you feel that this manuscript is now acceptable for publication, you may indicate that here to bypass the “Comments to the Author” section, enter your conflict of interest statement in the “Confidential to Editor” section, and submit your "Accept" recommendation.

Reviewer #1: All comments have been addressed

Reviewer #2: All comments have been addressed

2. Is the manuscript technically sound, and do the data support the conclusions?

Reviewer #1: Yes

Reviewer #2: Yes

3. Has the statistical analysis been performed appropriately and rigorously? 

Reviewer #1: Yes

Reviewer #2: Yes

4. Have the authors made all data underlying the findings in their manuscript fully available?

Reviewer #1: Yes

Reviewer #2: Yes

5. Is the manuscript presented in an intelligible fashion and written in standard English?

Reviewer #1: Yes

Reviewer #2: Yes

6. Review Comments to the Author

Reviewer #1: Thank you for giving me the chance to read this manuscript and for addressing adequately all the raised queries. I do not have anything more to add.

Reviewer #2: (No Response)

7. PLOS authors have the option to publish the peer review history of their article (what does this mean?). If published, this will include your full peer review and any attached files.

Reviewer #1: No

Reviewer #2: No

---

## [Editor Report · Acceptance letter]

8 Apr 2021

PONE-D-20-32723R1 

A dose-dependent beneficial effect of methotrexate on the risk of interstitial lung disease in rheumatoid arthritis patients 

Dear Dr. Kur-Zalewska:

I'm pleased to inform you that your manuscript has been deemed suitable for publication in PLOS ONE. Congratulations! Your manuscript is now with our production department. 

Kind regards, 

on behalf of

Dr. Minghua Wu 

Academic Editor

PLOS ONE